# Athletes’ Sensory Evaluation and Willingness to Pay for High-Protein Bread

**DOI:** 10.3390/foods14152673

**Published:** 2025-07-29

**Authors:** Roberta Selvaggi, Matilde Reitano, Elena Arena, Antonia Grasso, Biagio Pecorino, Gioacchino Pappalardo

**Affiliations:** 1Department of Agriculture, Food and Environment (Di3A), Agricultural Economics and Valuation Section, University of Catania, Via Santa Sofia n. 98-100, 95123 Catania, Italy; roberta.selvaggi@unict.it (R.S.); biagio.pecorino@unict.it (B.P.); gioacchino.pappalardo@unict.it (G.P.); 2Department of Agriculture, Food and Environment (Di3A), Food Technology Section, University of Catania, Via Santa Sofia n. 98-100, 95123 Catania, Italy; earena@unict.it (E.A.); antonia.grasso@unict.it (A.G.)

**Keywords:** functional food, sensorial analysis, willingness to pay, experimental auction, consumer behavior, food perception

## Abstract

The intrinsic relationship between food and health has led to growing interest in functional foods, particularly among athletes seeking to optimize performance and recovery. This study investigates the impact of product information and sensory attributes on athletes’ willingness to pay for an innovative high-protein bread. Utilizing a two-treatment experimental design, athletes were exposed to sensory evaluations either before or after receiving information. A combination of hedonic sensory analysis and economic evaluation assessed preferences through a non-hypothetical auction. Findings show that both sensory attributes—especially taste and aroma—and product information significantly influenced willingness to pay. The order of presentation played a crucial role: providing information first enhanced perceived value more strongly. While sensory evaluation moderately increased willingness to pay, product information had a stronger impact. A key contribution of this study is its novel evidence on how athletes balance sensory and informational cues in food evaluation—an aspect rarely explored. Contrary to assumptions that athletes ignore sensory quality due to their focus on nutrition, they did value sensory aspects, though they prioritized product information. These findings suggest that developing functional foods for athletes should integrate nutritional benefits and sensory appeal, as both elements contribute to acceptance and potential market success.

## 1. Introduction

The intrinsic link between food and health has always been recognized, with the understanding that what we eat has a significant impact on our physical and mental well-being [1]. Food has always played a key role, and over the years, there has been an increasing development, not only in the basic concept of food, but also in the attention given to its quality and safety [2,3,4].

In the light of this growing awareness of the function of food in the early 1980s, the link between certain food components and certain biologically active components that improve health status, reduce the occurrence of diseases, and improve immunological defenses, cellular functions, metabolic processes of the body, etc., was developed. In fact, it was defined that a food can be classified as functional if it demonstrates the ability to positively influence specific functions of the body that go beyond simple basic nutritional effects, such that the food is significant in improving the overall health status and well-being of the individual or in reducing the risk of disease [5].

Subsequently, the term functional food was born in Japan in 1988 as part of the Systemic Analysis and Development of Food Functions project, specifically to refer to nutrient-rich foods with beneficial health effects. The term then spread to various countries with different definitions [6]. In Europe, the real interest in the innovation and evolution of functional foods only began in the 21st century with EC Regulation 1924/2006 [7] and with the changing perception and purchasing choices of consumers, who are becoming increasingly food-conscious.

To date, among the many trends emerging in the food context, there is a growing interest in foods that combine taste, convenience, and an optimal nutritional profile [8,9]. In response to this growing demand, the food industry is constantly innovating and developing products that offer advanced nutritional solutions without compromising taste [10,11,12,13]. In particular, there has been a growing awareness among athletes about the importance of incorporating adequate amounts of protein into the daily diet to support overall health, optimize physical performance, and meet their high energy needs [14,15]. Athletes typically require specific dietary strategies that consider their unique energy needs and recovery requirements [16]. The role of functional foods becomes particularly relevant here as these foods can prevent chronic inflammation and support recovery from training-induced stress [17].

In this context, bread, a mainstay of the human diet for centuries, has been reinvented to meet modern nutritional needs [18]. The addition of high-quality protein to bread formulations offers a unique opportunity to increase the protein content of foods consumed daily [19,20], providing an important contribution to the protein diet of physically active and athletic individuals, supporting muscle protein synthesis and contributing to recovery after training and competition [21,22]. Therefore, in the context of a research project funded by the European Commission, the formulation of an innovative high-protein bread has emerged as a promising solution to meet the growing demand for healthy and functional foods for athletes.

The innovative bread was formulated with the aim of providing a high protein content while maintaining an acceptable appearance, taste, and texture. To achieve this goal, given the positive findings found in the literature [23,24,25,26], the bread formulation included plant protein sources such as chickpea flour (*Cicer arietinum*), selected for its high protein content (22%) and its amino acid profile complementary to wheat flour. Soy protein, known for its water absorption capacity and favorable nutritional profile, was also added. In addition, whey protein was incorporated due to its excellent techno-functional properties (particularly its gelling capacity) and its rich amino acid composition, being especially rich in lysine, methionine, and tryptophan [27]. Using both protein sources softens the dough due to the addition of milk protein, which decreases dough stability by weakening the gluten network. Meanwhile, soy protein enhances dough stability and elasticity due to protein aggregation [28].

Beyond the protein sources, sesame, sunflower, and flaxseeds, which are rich in plant-based omega-3 polyunsaturated fatty acids known for their health benefits, were included [29,30].

However, the introduction of a high-protein bread raises a number of scientific and practical questions that need to be thoroughly investigated. Reitano et al. (2024) [31] highlight how athletes’ preferences influence their WTP for such a product, suggesting that a thorough understanding of these dynamics is critical. The literature has extensively documented that sensory attributes, such as appearance, color, odor, taste, and texture, are key factors influencing consumer preferences and acceptance of food products. This is particularly relevant for high-protein products, which often present sensory challenges, such as off-flavors or textural differences, due to the incorporation of alternative protein sources. It is known that legume proteins have a distinct “beany” flavor, which contrasts with the industry’s desire for its products to have a neutral taste [32].

Saint-Eve et al. (2018) [33] report that the appearance and texture of a high-protein snack influence preference, whereas pea and green flavor perceptions appear to be a barrier to acceptance; the higher the pea content, the less the snack was appreciated. At the same time, pulses and plant-based alternative proteins have the highest acceptance level with respect to other alternative proteins [34].

Therefore, consumer acceptance of these products strongly depends on the ability to maintain desirable sensory characteristics while meeting nutritional expectations [35,36,37]. Additionally, it is known that specific product information plays a significant role in shaping consumers’ perceptions and purchase decisions [38,39]. Other studies have examined the joint effect of information and sensory hedonic evaluation on willingness to pay (WTP) [40,41,42,43]; however, no study has examined this effect in a specific context, such as athletes.

Our study stands out for testing these aspects on a sample of athletes, examining whether information and sensory hedonic evaluation had an impact on their behavior, particularly on WTP for a high-protein bread. Based on the existing literature, we hypothesize that both information provision and sensory hedonic evaluation significantly affect athletes’ WTP, with information potentially enhancing product appreciation and WTP more than sensory attributes alone. We also explored whether the predisposition of athletes to sacrifice the sensory aspect for performance is really valid and whether the effect of the ordering of information and sensory hedonic evaluation could influence participants’ evaluations. To achieve those goals, the specific research questions in this study are as follows:-Research question 1 (RQ1): What is the impact of information presentation on WTP for high-protein bread?-Research question 2 (RQ2): How do sensory evaluations of high-protein bread affect athletes’ WTP?-Research question 3 (RQ3): Does the order in which information and sensory hedonic evaluation of high-protein bread are presented have a significant effect on evaluations and WTP?-Research question 4 (RQ4): Which sensory attributes are most correlated with WTP for high-protein bread among athletes?

To address these research questions, we employed an experimental second-price auction as a methodological tool to assess athletes’ WTP, complemented by a sensory hedonic evaluation phase that constituted one of the experimental rounds. This dual approach allows the athletes’ sensory preferences to be explored in a detailed and accurate manner [44], providing an in-depth understanding of the organoleptic characteristics of high-protein bread and its potential implications on consumer satisfaction and purchasing habits.

By integrating sensory hedonic evaluation with economic analysis, this study offers a comprehensive perspective on athletes’ attitudes towards high-protein bread. This approach provides valuable recommendations for the development and marketing of innovative food products aimed at this specific consumer segment, emphasizing the importance of considering both sensory and economic aspects in an integrated way.

## 2. Materials and Methods

### 2.1. Data Collection

From May to October 2022, an experimental study involving athletes was conducted in Sicily (Italy), with the aim of estimating the athletes’ liking and WTP for a high-protein functional food, using a high-protein bread as a case study.

Considering the nature of the product and the target consumers for the product under study, the selection of participants was performed in order to include only athletes recruited from specific sports centers (gyms and swimming pools) and not to include subjects with intolerance or allergy to one or more of the ingredients, who would not be able to undergo the sensory hedonic evaluation. Selected athletes were invited to report to the experimental laboratory of the local university at agreed-upon dates and times.

After 521 potential participants were contacted and 333 appointments were arranged, 189 valid observations were collected at the end of the survey. This corresponds to a participation rate of 57%.

The athletes involved were invited to participate in an experimental auction where they were asked to bid the maximum price they would be willing to pay for a 100 g pack of high-protein bread. The auction was designed to assess how specific information about the product and sensory hedonic evaluation (color, odor, taste, texture, and overall liking degree) impacted their WTP, thereby providing insights into the factors that influence their purchasing decisions. Sensory evaluation was conducted under controlled conditions (22 °C ± 1 °C, RH 45–55%); the room was quiet and well lit. The sample of high-protein bread was served on a white plate at room temperature, and participants were instructed to clean their palate with water before tasting. Because only one sample was evaluated, no coding procedures were required.

Participants agreed to take part in the research, and written informed consent was obtained from all subjects in line with the General Data Protection Regulation (GDPR) 2016/679. This study was conducted according to the principles established in the Declaration of Helsinki, as our institution does not have an ethics committee for sensory and food quality evaluation studies.

### 2.2. High-Protein Bread Production

In collaboration with a bread-making company (Cooperativa Valle del Dittaino, Enna, Italy), experimental bakery tests were conducted to optimize the formulation and produce bread with acceptable physical and sensory characteristics. High-protein bread used was produced according to the formulation reported in Table 1.

Dough preparation: the ingredients were mixed for 6 min in a high-speed mixer with variable speed (San Cassiano, Italy). After 10 min, the dough was formed into portions of about 56–58 g, and the loaves were leavened for 90 min at 32 °C and 87% humidity (Saccani, Italy). The loaves were baked at 280 °C for 8 min, in an industrial tunnel oven (Werner, Italy). The loaves were, then, automatically transported into the cooling chamber (room temperature) for 20 min. After cooling, the loaves were surface-treated with ethyl alcohol and packed in flow packs. Figure 1 shows the final product.

### 2.3. Auction Mechanism

A second-price experimental auction was performed to assess athletes’ WTP for high-protein bread. This auction mechanism was selected because it is widely recognized as an incentive-compatible method that elicits truthful bidding behavior, as participants are motivated to reveal their true valuation of the product [45,46]. This approach has been extensively used in consumer preference studies for food products due to its ability to provide a direct, quantitative measure of individuals’ WTP [47,48].

The experimental design allowed the evaluation of athletes’ WTP under two different treatments: one focused on the impact of providing information about the high-protein bread (Treatment 1—T1), and the other on the effect of liking degree on participants’ WTP (Treatment 2—T2). The 189 participants were evenly divided between the two treatments, with a total of 98 participants for T1 and 91 for T2.

By combining sensory hedonic evaluation with the second-price auction, the method provides a comprehensive assessment of the value athletes attribute to the product’s nutritional and sensory characteristics. This approach not only facilitates the estimation of WTP in an incentivized context but also aligns with previous studies that have used auction methods to evaluate consumer preferences for innovative food products [49,50].

As shown in Table 2, for each treatment, each participant performed 3 auction rounds, and participation in all rounds was mandatory for all subjects. However, to minimize anchoring effects or influence from previous bids, participants were not provided with any feedback about their WTP results between rounds.

For both treatments, in the first round (R1), participants were introduced to the experimental high-protein bread without any specific information about the product, but only general information, i.e., that the subject of the auction would be two 50 g breads inside the 100 g package and that their characteristic feature was the high protein content guaranteed by the addition of chickpea flour, a source of vegetable protein. This baseline round was defined as the “No-info” treatment. They were then asked about their WTP for the experimental high-protein bread.

In the second round (R2) of Treatment 1, participants were given detailed information about the high-protein bread before the WTP evaluation. This treatment was defined as “Info about protein bread”.

Specifically, specific product information was provided by showing the following text:

“Proposed in practical packages of two 50 g ready-to-eat high-protein breads, obtained using wheat and legumes. It is a healthy product and it is ideal for those who want to increase their daily protein intake, having to give up a food that is traditionally present in the diet, promoting muscle growth or maintenance for those who lead an active lifestyle, practice sport or want to increase muscle mass.

The claim “high in protein” is applied to the product because more than 20% of the energy value of the food is provided by protein (EC Reg. 1924/2006 [7]).

100 g of Conventional Bread → 8.6 g of protein!

100 g High-Protein Bread → 20 g protein

The World Health Organization (WHO) has published population reference intake for protein.

For an 80 kg athletic person doing aerobic activity, the daily protein requirement is 80 g. So, 100 g of high-protein bread = 25% of the daily requirement”.

In the third round (R3) of Treatment 1, a sensory hedonic evaluation session of the experimental high-protein bread was conducted before the WTP evaluation. The assessment focused on the evaluation of color, odor, taste, texture, and overall liking degree of the high-protein bread sample using a 9-point hedonic scale (1 = extremely unpleasant, 5 = neither unpleasant nor pleasant, 9 = extremely pleasant) [51]. This treatment was defined as “sensory hedonic evaluation”. This made it possible to evaluate not only the product liking degree, but also the variability of WTP following the sensory hedonic evaluation of the product.

The variation between the first and second treatments consisted of reversing the order of rounds 2 and 3. In Treatment 2, the sensory hedonic evaluation was performed before the detailed information about the high-protein bread was provided.

In each round, participants had the opportunity to participate in the experimental second-price auction, in which they had to indicate the maximum price they would be willing to pay for the experimental high-protein bread, thus providing a direct measure of their WTP for the product. This method of revealed preferences is particularly advantageous as it captures authentic consumer behavior in a realistic purchasing context [52]. Unlike hypothetical surveys that may be influenced by different biases or a lack of engagement, the second-price auction incentivizes participants to consider their true valuation of the product [53].

This methodological approach allows us to examine the impact of information and sensory hedonic evaluation on participants’ purchase behavior in detail, providing valuable information for understanding the dynamics that drive consumers’ purchase decisions towards high-protein bread.

At the conclusion of each experimental auction, participants were administered a comprehensive questionnaire aimed at profiling consumer characteristics. The questionnaire collected data on the following aspects: socio-demographic information, including gender, age, education level and income; the level and duration of physical activity undertaken by the participants; whether participants followed a dietary plan prescribed by a nutritionist or specialist. This allowed for a detailed analysis of consumer profiles.

### 2.4. Experimental Auction Procedure

Auction participants were welcomed to the experimental laboratory at the agreed-upon time and date, where they were randomly and anonymously assigned an ID in accordance with data protection regulations. Each auction session included a maximum of 10 participants, divided into subgroups of 4 or 5. In total, 189 athletes participated in the experimental auction, which consisted of 23 separate sessions.

The experimental auction began with a three-round training auction using a test product, a 500 g packet of pasta, to ensure that all participants understood the bidding mechanism.

Following this trial session, participants took part in the actual auction, where the high-protein bread (presented as a 100 g packet) was evaluated over three rounds.

After each round, the participants bid for the auction item. At the end of the three bidding rounds, one of the rounds was randomly selected, and the participant with the highest bid in that round had the opportunity to purchase the product at the second-highest price offered in the same round, rather than at the price he or she had bid, through a monetary transaction.

At the end of the experimental procedure, each participant received an EUR 15 gadget box as a token of appreciation for their participation in the study.

### 2.5. Data Processing

The data processing phase utilized Stata SE^®^ software version 17.0 (Stata Corp LLC, College Station, TX, USA) to perform statistical analyses, including *t*-tests and regression analysis, aimed at evaluating the WTP among participants under different treatment conditions across multiple rounds. To assess the impact of treatments on WTP, *t*-tests were conducted for within-group analysis of each treatment group, comparing WTP values between successive rounds.

Moreover, a multiple regression analysis was conducted to further investigate the factors influencing changes in WTP. The regression model incorporated independent variables representing treatment type and sensory hedonic evaluation of each attribute (color, flavor, taste, texture, and overall liking). The analysis aimed to quantify the impact of these variables on WTP, assessing which specific factors had a statistically significant effect on athletes’ WTP.

For the sensory hedonic evaluation, the average score for each evaluated attribute of the product was used.

To enhance the clarity of the analysis performed, Table 3 provides the coding details for the key variables used in the study.

## 3. Results

### 3.1. Sample Characteristics

Summary statistics of the participants are shown in Table 4. The sample consists of 189 participants with an average age of 35 years and a slight predominance of males compared to females. The relatively young age of the sample is an important finding as it may have influenced the positive acceptance of high-protein bread, as younger consumers tend to be more open to trying innovative and functional food products. The majority of participants had a net annual household income between EUR 20,000 and EUR 39,999. Educationally, the participants are primarily high school graduates and university degree holders. The activity level is mostly amateur, with fewer participants involved at competitive or elite levels. Regarding dietary habits, a quarter of the sample follows a specific food plan, while the majority does not.

### 3.2. Sensory Analysis

Table 5 presents the sensory hedonic evaluation of a high-protein bread, evaluated on a scale from 1 to 9 for five attributes: color, odor, taste, texture, and overall bread liking.

The analysis of the sensory hedonic evaluation results highlights notable differences in how the high-protein bread’s attributes were perceived under the two experimental conditions (Treatment 1 and Treatment 2). Color, which received the highest average rating among all attributes, was evaluated significantly more positively by participants in Treatment 1 (8 ± 1) than those in Treatment 2 (7 ± 1), with the difference being statistically significant (*p* < 0.05). This suggests that the visual appearance of the experimental high-protein bread was more appealing under the first condition, potentially influenced by contextual or informational factors. Regarding odor, the overall average rating was slightly lower at 6, indicating a good but less enthusiastic appreciation of the high-protein bread’s smell. When analyzed by treatment, odor scores were again higher in Treatment 1 (7 ± 2) compared to Treatment 2 (6 ± 2), further emphasizing a perceptible difference between the two conditions. Taste ratings followed a similar trend to odor, with an overall average of 6. Participants in Treatment 1 gave higher scores (7 ± 1) than those in Treatment 2 (6 ± 2), suggesting that flavor perception was positively influenced by the information provided in the first treatment. For texture, the average score was 7 in both treatments, although participants in Treatment 2 rated it slightly higher (7 ± 1) than those in Treatment 1 (7 ± 2). Finally, the overall liking degree of the experimental high-protein bread received relatively high ratings, with an average score of 7. However, participants in Treatment 1 reported significantly higher liking (7 ± 1) compared to those in Treatment 2 (6 ± 1). This suggests that the first treatment condition created a more favorable impression of the product as a whole.

It is also important to note that for all attributes, there was considerable variability in individual scores, as reflected by the standard deviations reported for each treatment group. This variability likely reflects individual differences in preferences among evaluators and highlights the importance of considering both group-level trends and individual-level variability when interpreting sensory data.

In summary, while participants generally appreciated the experimental bread across all attributes, significant differences emerged between treatments for most sensory characteristics except texture. These findings underscore the importance of information in shaping sensory perceptions.

### 3.3. Willingness to Pay

Table 6 shows the mean values for WTP across the four treatments and for each round of the experimental auction.

To evaluate the impact of the information treatment and sensory hedonic evaluation on participants’ WTP for experimental high-protein bread across the two treatments (T1 and T2) and for different rounds (R1, R2, and R3), a *t*-test was conducted, as reported in Table 7.

The *t*-test results show that in T1, the introduction of information in R2 had a significant and positive impact on participants’ WTP, with a significant increase in mean bid from R1 to R2 *(p* < 0.0001). However, sensory experience in the R3 did not produce a significant change in WTP compared with the information phase of R2 (*p* = 0.3197). In T2, there is a slight but significant increase in WTP from R1 to R2 (*p* = 0.0417), indicating a positive effect of sensory experience. Subsequently, the addition of information further significantly increased WTP (*p* < 0.0001), showing that although sensory experience has an initial impact, information is the most influential factor in determining the perceived value of the product.

To estimate the factors influencing consumers’ WTP, a linear regression analysis was performed (Table 8). Specifically, it was conducted using the variable measuring the change in WTP between treatment rounds as the dependent variable. The regression model is specified asDeltaBid = β_0_ + β_1_ × Treatment + β_2_ × Color + β_3_ × Flavor + β_4_ × Taste + β_5_ × Texture + ε where DeltaBid is the dependent variable, indicating the difference in bids made by participants between the two experimental treatments (the objective is to understand what influences this variation). Specifically, DeltaBid is calculated as the difference between the R3 bid and R2 bid for T1 participants and as the difference between the R2 bid and R1 bid for T2 participants. This distinction ensures that the analysis captures the variation in bidding behavior resulting from the sequential exposure to rounds across different treatments. Coefficient β_0_ is the constant term of the regression and represents the average value of DeltaBid when all other independent variables (treatment, color, flavor, taste, texture) are zero. β_1_, β_2_, β_3_, β_4_, and β_5_ are the independent variables representing the type of treatment carried out (Treatment 1 and Treatment 2) and the score participants gave to the various aspects of sensory hedonic evaluation, i.e., the color, flavor, taste, and texture of the high-protein bread. The term ε represents the variability not explained by the model.

These results suggest that the change in WTP is influenced by the treatment and the participant’s liking degree for the experimental high-protein bread’s odor and taste.

In particular, results from the regression model imply that the type of treatment has a significant effect on the change in WTP. In fact, the coefficient for treatment is positive and significant (β = 0.16, *p* < 0.01), indicating that participants assigned to T2 exhibit a higher mean WTP compared to those in T1. This suggests that the type of treatment has a positive effect on WTP, as it increased the mean WTP by EUR 0.16 between rounds. The model explains 20.7% of the variance in WTP (R^2^ = 0.207, F(5, 183) = 9.56, *p* < 0.001). Among the sensory variables, flavor and taste were significantly correlated with the change in WTP. Flavor had a positive β of 0.0397 (*p* = 0.040). This means that an increase of 1 unit in taste appreciation is associated with an average increase in WTP of 0.040 EUR/packet. Taste has a positive coefficient of 0.0747 (*p* < 0.001), meaning that a 1-unit increase in taste rating is associated with an average increase of 0.075 EUR/packet in WTP. In contrast, the variables color (β = −0.214; *p* = 0.344) and texture (β = 0.0043, *p* = 0.824) show no significant effect.

## 4. Discussion

The research examined two main aspects: the importance of the sensory hedonic evaluation of high-protein bread among athletes and the impact of information on their WTP. Initial hypotheses suggested that the order of information presentation, both before and after sensory hedonic evaluation, could significantly influence athletes’ WTP. By addressing the four key research questions, the study provides a comprehensive understanding of athletes’ WTP for high-protein bread, highlighting the effect of the order of presentation of sensory hedonic evaluation and information.

-RQ1: What is the impact of information presentation on WTP for high-protein bread?

The results of the second-price auction show that the provision of information significantly influences athletes’ WTP. In T1, a significant increase in WTP was observed between the first bidding round, where no information was provided, and the second bidding round, where information about high-protein bread was provided. Similarly, in T2, where information was provided after sensory hedonic evaluation, WTP increased further in the third round. This finding highlights the pivotal role of information in shaping consumers’ valuation of food products, particularly those with specific nutritional claims. These results align with previous studies showing that detailed product information, especially regarding nutritional benefits, enhances consumers’ perceived value and WTP [54,55,56,57]. However, the novelty of this study lies in demonstrating that this effect was also present in athletes.

-RQ2: How do sensory hedonic evaluations of high-protein bread affect athletes’ WTP?

Sensory hedonic evaluation also played a crucial role in determining WTP. In T2, where sensory evaluation preceded the provision of information, a significant increase in WTP was recorded from the first to the second round of bids. This suggests that sensory experiences can enhance athletes’ perception of the product’s value. However, the subsequent provision of information further increased WTP, underscoring that while sensory hedonic evaluation is important, information remains the dominant factor in shaping consumers’ overall valuation of the product. This result confirms what has been reported in the literature that sensory attributes can influence consumers’ purchasing decisions [35,36,37,58]. Interestingly, the results suggest that although athletes are more health-conscious consumers, their purchasing behavior follows similar patterns to that of the general population.

-RQ3: Does the order in which information and sensory evaluation of high-protein bread are presented have a significant effect on evaluations and WTP?

The findings suggest that the order of information and sensory hedonic evaluation significantly impacts WTP. In both treatments, an increase in WTP was observed after participants received additional information or completed the sensory evaluation. However, differences emerged in the magnitude of these changes. In T1, where information was provided before the sensory evaluation, WTP increased significantly from R1 to R2, demonstrating the immediate effect of nutritional information on perceived value. The subsequent sensory evaluation (R3) led to a slight decrease in WTP, suggesting that sensory impressions tempered the initial expectations formed by the information. Conversely, in T2, where sensory evaluation preceded the provision of information, WTP grew more gradually from R1 to R2, indicating a moderate effect of sensory experience alone. However, when nutritional information was introduced in R3, WTP increased further, demonstrating that information played a crucial role in enhancing perceived value after the product had been tasted. These results suggest that presenting information first (T1) creates an initial surge in WTP, which is then adjusted based on sensory feedback, while presenting sensory evaluation first (T2) results in a more progressive increase, with the strongest effect occurring when information is introduced. These findings support the hypothesis that prior information positively influences sensory perceptions by creating expectations that enhance sensory experiences [41,59]. This study adds novel evidence demonstrating that, among athletes, information plays a more decisive role than sensory hedonic evaluation in shaping product valuation.

-RQ4: Which sensory attributes are most correlated with WTP for high-protein bread among athletes?

Regression analyses reveal that among the sensory attributes, taste and aroma are the most significant predictors of WTP. These results are consistent with previous research showing that taste and aroma are primary drivers of food product preferences [60,61]. The significant correlation between these sensory attributes and WTP emphasizes the importance of enhancing these characteristics in product development to increase consumer acceptance. This finding highlights that, despite athletes’ strong focus on nutritional attributes, sensory enjoyment remains a critical factor in food choice.

The results obtained in this study are in line with the observations of Reitano et al. (2024) [31], which indicate that athletes show a greater WTP for food products that meet specific nutritional needs. A key contribution of this study is that it provides empirical evidence on how athletes balance sensory attributes and product information in their purchasing decisions, an aspect that has never been explored in the previous literature. While athletes’ purchasing behavior follows similar patterns to that of general consumers, their decision-making process places greater emphasis on nutritional information over sensory attributes. Interestingly, this finding contrasts with the common assumption that athletes, due to their functional approach to nutrition, would be largely indifferent to sensory characteristics. Instead, our results show that while they prioritize product information, sensory attributes still play a role, albeit a secondary one. Unlike the general population, for whom sensory characteristics often play a dominant role in food choices, athletes appear more willing to compromise on sensory enjoyment in favor of nutritional benefits. These insights offer valuable implications for the development and marketing of functional foods aimed at this specific target group.

## 5. Conclusions

In conclusion, the study confirmed the initial hypotheses regarding the importance of information and sensory preferences in determining WTP for high-protein bread. The results clearly indicate that while sensory hedonic evaluation plays a role in shaping WTP, information exerts the strongest influence on perceived value.

The practical implications of these findings can guide development and marketing strategies for similar food products, thus helping to improve the acceptance and commercial success of high-protein products. An effective marketing strategy should include clear and detailed product information while also emphasizing key sensory attributes valued by consumers.

Future studies could expand the research by including a more diverse sample of consumers and analyzing different types of food products to assess whether these effects hold across broader contexts. Additionally, exploring the effect of different methods of presenting information and sensory experiences could help optimize marketing and communication strategies to enhance consumers’ perception and WTP.

## Figures and Tables

**Figure 1 foods-14-02673-f001:**
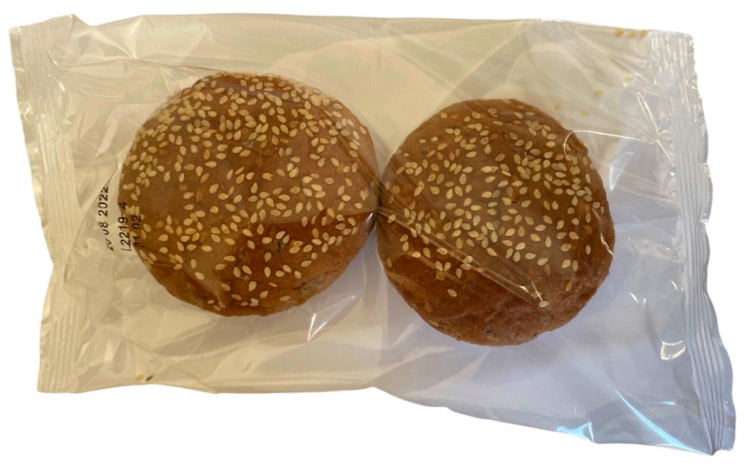
Package of experimental high-protein bread (100 g).

**Table 1 foods-14-02673-t001:** High-protein bread formulation.

Ingredient	Quantity (%)
Water	31.89
Wheat flour	12.75
Blend of cereal flour	11.39
Blend of seeds	11.39
Wheat gluten	11.39
Chickpea flour	4.33
Milk protein	2.85
Yeast	4.78
Skim milk powder	2.28
Sunflower seed oil	2.05
Soy protein	1.37
Salt	1.03

**Table 2 foods-14-02673-t002:** Experimental design.

Round	Treatment 1	Treatment 2
R1	No-info	No-info
	WTP for experimental bread	WTP for experimental bread
R2	Info about protein bread	Sensory hedonic evaluation
	WTP for experimental bread	WTP for experimental bread
R3	Sensory hedonic evaluation	Info about protein bread
	WTP for experimental bread	WTP for experimental bread

**Table 3 foods-14-02673-t003:** Variable coding and descriptions.

Variable Description	Possible Values	Type
Willingness to pay	Bid in euros for 1 package of high-protein bread (100 g)	Continued
Gender	1 = male 2 = female	Dummy
Age	Number of years	Continued
Education level	1 = middle school 2 = high school 3 = university degree 4 = postgraduate	Categorial
Annual net household income	1 = less than EUR 20,000 2 = from EUR 20,000 to EUR 39,000 3 = from EUR 40,000 to EUR 59,000 4 = from EUR 60,000 to EUR 79,000 5 = over EUR 100,000	Categorial
Years of sports practice	1 = less than 1 year 2 = from 1 to 3 years 3 = over 3 years	Categorial
Level of activity	1 = amateur 2 = competitive 3 = elite	Categorial
Follows an eating plan	1 = yes 2 = no	Dummy
How much do you like the color	1 = extremely unpleasant 2 = very unpleasant 3 = unpleasant 4 = slightly unpleasant 5 = neither unpleasant nor pleasant 6 = slightly pleasant 7 = pleasant 8 = very pleasant 9 = extremely pleasant	Categorial
How much do you like the smell	1 = extremely unpleasant 2 = very unpleasant 3 = unpleasant 4 = slightly unpleasant 5 = neither unpleasant nor pleasant 6 = slightly pleasant 7 = pleasant 8 = very pleasant 9 = extremely pleasant	Categorial
How much do you like the taste	1 = extremely unpleasant 2 = very unpleasant 3 = unpleasant 4 = slightly unpleasant 5 = neither unpleasant nor pleasant 6 = slightly pleasant 7 = pleasant 8 = very pleasant 9 = extremely pleasant	Categorial
How much do you like the texture	1 = extremely unpleasant 2 = very unpleasant 3 = unpleasant 4 = slightly unpleasant 5 = neither unpleasant nor pleasant 6 = slightly pleasant 7 = pleasant 8 = very pleasant 9 = extremely pleasant	Categorial
How much do you like the high-protein bread	1 = extremely unpleasant 2 = very unpleasant 3 = unpleasant 4 = slightly unpleasant 5 = neither unpleasant nor pleasant 6 = slightly pleasant 7 = pleasant 8 = very pleasant 9 = extremely pleasant	Categorial

**Table 4 foods-14-02673-t004:** Characteristics of the sample.

Variable	n.	%
*Gender*		
Male	111	58.72%
Female	77	40.74%
I prefer not to answer	1	0.54%
*Age*		
18–35	116	61.38%
36–50	46	24.34%
over 50	27	14.28%
*Net annual household income*		
Less than EUR 20,000	56	29.63%
Between EUR 20,000 and EUR 39,999	80	42.33%
Between EUR 40,000 and EUR 59,999	31	16.40%
Between EUR 60,000 and EUR 79,999	13	6.88%
More than EUR 80,000	7	3.70%
I prefer not to answer	2	1.06%
*Level of education*		
Secondary school	2	1.06%
High school	83	43.92%
University degree	78	41.27%
Postgraduate	26	13.75%
*How long have you been practicing sport*		
Less than 1 year	32	16.93%
From 1 to 3 years	27	14.29%
Over 3 years	130	68.78%
*Activity level*		
Amateur	140	74.07%
Competitive	30	15.87%
Elite	19	10.05%
*Following a food plan*		
Yes	46	24.34%
No	143	75.66%

**Table 5 foods-14-02673-t005:** Sensory hedonic evaluation of experimental high-protein bread.

	Treatment 1	Treatment 2
Color	7.51 ± 1.04 a	6.79 ± 1.33 b
Odor	6.63 ± 1.52 a	6.00 ± 1.56 b
Taste	6.62 ± 1.41 a	5.83 ± 1.90 b
Texture	6.73 ± 1.54 a	6.98 ± 1.19 a
Liking degree	6.99 ± 1.32 a	6.21 ± 1.38 b

Mean value ± standard deviation. Different letters in the same row indicate a significant difference between the mean values (*p* < 0.05).

**Table 6 foods-14-02673-t006:** Mean bid values of all groups.

	T1 (98 Units)		T2 (91 Units)	
	Mean WTP [EUR]	Standard Deviation	Mean WTP [EUR]	Standard Deviation
R1	1.70	1.07	1.34	0.65
R2	1.89	1.06	1.41	0.70
R3	1.85	1.10	1.53	0.73

**Table 7 foods-14-02673-t007:** *t*-test for equality of mean WTP within groups.

	T1 (98 Units)		T2 (91 Units)	
	Mean WTP Differences	*p*-Value	Mean WTP Differences	*p*-Value
R1 vs. R2	−0.18	0.0000 ***	−0.07	0.0417 **
R2 vs. R3	0.04	0.3197	−0.12	0.0000 ***

Note: The null hypothesis is mean difference = 0, and the probability is (|T| > |t|). **, and *** denote significance at 5%, and 1% levels, respectively.

**Table 8 foods-14-02673-t008:** Regression analysis of factors influencing WTP.

Variable	Coefficient (β)	*p*-Value
Treatment	0.160	0.002 ***
Color	−0.214	3.444
Odor	0.040	0.040 **
Taste	0.075	0.000 ***
Texture	0.004	0.824
Cons	−0.664	0.000 ***

**, and *** denote significance at 5%, and 1% levels, respectively. Model statistics: R-squared: 0.2071; adjusted R-squared: 0.1855.

## Data Availability

Data will be made available upon request.

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
