# Peer review of "Athletes’ Sensory Evaluation and Willingness to Pay for High-Protein Bread"

_foods, 2025, doi:10.3390/foods14152673_

Round 1
Reviewer 1 Report
Comments and Suggestions for Authors
This manuscript has an interesting and relevant topic but still needs some work to improve the quality of presentation and depth of analysis.
- The Introductionpart explains a lot of general background about food and health,but lack of the detailed overview of sensory evaluation and consumer acceptance of high protein products.
- Line 74-75. “The innovative bread was formulated with the aim of providing a high protein content without compromising taste and texture.”Have the authors experimentally validated that the results from this manuscript meet this sentence?
- 2. High-protein bread production. There are different types of protein in the high-protein bread formulation, seems very complicated, how the authors decidedthis formulation? Why add milk protein and soy protein?
- 3. Auction mechanism. Is it required for every consumer to participate in all three rounds of the WTP experiment? Were the results of WTP in R2 and R3 influenced by the first WTP in R1?
- Please explain the reasons why use chickpea flour to increase protein content.
- Change 2.5 to 2.4. Change 2.6 to 2.5.
- Line 241-243. this sentence was confusing.
- Line 275. with an average of 34 and 96? And also consider the age on the influenceof the bread acceptance.
- 2 Sensory analysis. Please add the data of average score for the treatmentstogether.
- Line 300. What the authors mean by “ could benefit from refinement toachieve greater consistency across contexts.”?
- Why the texture presents a different pattern? please cite references to explainyour statement.
- RQ2: How do sensory hedonic evaluations affect athletes' WTP? Line413-414: This result is consistent with prior studies showing that sensory attributes have a direct influence on consumers' purchasing decisions. The experimental design of this manuscriptis completely different from those references mentioned. The experiments in this manuscript can only confirm that conducting sensory hedonic evaluation first affect the athletes’
Author Response
Comment 1: The Introduction part explains a lot of general background about food and health, but lack of the detailed overview of sensory evaluation and consumer acceptance of high protein products.
Response 1: Thank you for pointing this out, we have integrated the introduction with sensory evaluation and consumer acceptance of high protein products.
Comment 2: Line 74-75. “The innovative bread was formulated with the aim of providing a high protein content without compromising taste and texture.” Have the authors experimentally validated that the results from this manuscript meet this sentence?
Response 2: Thank you, we appreciate your comment. We revised the sentence hopefully to more accurately reflect the scope of the study. While the formulation aimed to increase protein content without compromising sensory quality, the evaluation of taste and texture was based on preliminary chemical and sensory evaluations conducted during product formulation and subsequent sensory evaluations by participants in the present study, which indicated generally positive ratings.
Comment 3: High-protein bread production. There are different types of protein in the high-protein bread formulation, seems very complicated, how the authors decided this formulation? Why add milk protein and soy protein?
Response 3: Thank you for your comment. During the experimental tests conducted to optimize the formulation, the use of both protein sources resulted in better results, the addition of milk protein and soy makes the dough softer and more elastic. In the paper we have revised the sentence to clarify the rationale behind the selection of the protein sources.
Comment 4: Auction mechanism. Is it required for every consumer to participate in all three rounds of the WTP experiment? Were the results of WTP in R2 and R3 influenced by the first WTP in R1?
Response 4: Thank you for your observation. We have clarified this aspect in the revised manuscript. Specifically, we now state that all participants were required to take part in all three auction rounds for each treatment. Moreover, we specified that to reduce the potential influencing effects of earlier bids on later bids, no feedback was provided on WTP results between rounds.
Comment 5: Please explain the reasons why use chickpea flour to increase protein content.
Response 5: Thank you for your comment. We have specified why chickpea flour increase protein content.
Comment 6: Change 2.5 to 2.4. Change 2.6 to 2.5.
Response 6: Thank you for pointing this out, and apologies for the oversight. We have corrected the numbering as suggested.
Comment 7: Line 241-243. this sentence was confusing.
Response 7: Thank you for your comment. We have clarified the sentence to improve its readability and avoid confusion.
Comment 8: Line 275. with an average of 34 and 96? And also consider the age on the influence of the bread acceptance.
Response 8: Thank you for the observation. We agree and we have discussed that the average age of 35 may have influenced the bread acceptance.
Comment 9: Sensory analysis. Please add the data of average score for the treatments together.
Response 9: Thank you for the comment. We’ve specified the total average score for the treatments together.
Comment 10: Line 300. What the authors mean by “could benefit from refinement to achieve greater consistency across contexts.”?
Response 10: Thank you for the comment. We have clarified the sentence.
Comment 11: Why the texture presents a different pattern? please cite references to explain your statement.
Response 11: We thank you for this insightful observation. The previous formulation may have been unclear and potentially misleading. We have revised the sentence to more accurately reflect the results.
Comment 12: RQ2: How do sensory hedonic evaluations affect athletes' WTP? Line413-414: This result is consistent with prior studies showing that sensory attributes have a direct influence on consumers' purchasing decisions. The experimental design of this manuscript is completely different from those references mentioned. The experiments in this manuscript can only confirm that conducting sensory hedonic evaluation first affect the athletes’.
Response 12: We thank you for your remark. We have modified the paragraph to clarify that our results simply confirm what has been reported in the literature about recognizing the role of sensory experience in influencing consumer evaluation. We hope the revised version better reflects this nuance and avoids overgeneralization.

Reviewer 2 Report
Comments and Suggestions for Authors
The paper “Athletes’ sensory evaluation and willingness to pay for high-protein bread” contributes to the growing literature on research in the area of consumer behaviour towards functional foods, especially among athletes.
The work makes a significant contribution to the field.
The work is well organised and comprehensively described.
The work is scientifically sound and not misleading.
However, the following items should be revised:
Introduction
Line 83-85
“The literature has extensively documented that sensory attributes, such as appearance, color, odor, taste, and texture, are key factors influencing consumer preferences and acceptance of food products [29-31]” - Do the indicated literature references (29-31) apply only to bread? If so, I suggest replacing the words "food products" with "high-protein bread".
Line 101-105
The sensory evaluation concerns a specific product, so I suggest correcting the hypotheses to:
How do sensory evaluations affect athletes' WTP" to "...sensory evaluations of ..."
Similarly, the remaining ones, because the participants only evaluated high-protein bread
Materials and Methods
The authors did not describe the conditions under which the sensory evaluation was conducted. Similarly, how were the samples coded, and was the order of sample presentation random?
Results
Line 289-290
“Starting with color, participants in Treatment 1 rated the color significantly higher (7.51 ± 1.04) compared to those in Treatment 2 (6.79 ± 1.33).” Are these differences statistically significant? The authors write about this later, but it is misleading here.
Additionally, shouldn't the results be rounded to the values used on the scale? The scale is structured. Similarly, in the subsequent sensory analysis results
Line 320 – “p “ - it should be Italic (p), similar to others,
Author Response
Introduction
Comment 1: Line 83-85 “The literature has extensively documented that sensory attributes, such as appearance, color, odor, taste, and texture, are key factors influencing consumer preferences and acceptance of food products [29-31]” - Do the indicated literature references (29-31) apply only to bread? If so, I suggest replacing the words "food products" with "high-protein bread".
Response 1: Thank you for your comment, we appreciate your attention to precision. However, we would like to clarify that the indicated literature references (29–31) do not apply exclusively to bread but refer more broadly to food products.
Comment 2: Line 101-105 The sensory evaluation concerns a specific product, so I suggest correcting the hypotheses to: How do sensory evaluations affect athletes' WTP" to "...sensory evaluations of ..."
Similarly, the remaining ones, because the participants only evaluated high-protein bread.
Response 2: Thank you for pointing this out, we have adjusted everything.
Materials and Methods
Comment 3: The authors did not describe the conditions under which the sensory evaluation was conducted. Similarly, how were the samples coded, and was the order of sample presentation random?
Response 3: Thank you for the comment. We have clarified we have now included a detailed description of the sensory evaluation procedure in the paragraph 2.1.
Results
Comment 4: Line 289-290 “Starting with color, participants in Treatment 1 rated the color significantly higher (7.51 ± 1.04) compared to those in Treatment 2 (6.79 ± 1.33).” Are these differences statistically significant? The authors write about this later, but it is misleading here.
Additionally, shouldn't the results be rounded to the values used on the scale? The scale is structured. Similarly, in the subsequent sensory analysis results
Line 320 – “p “ - it should be Italic (p), similar to others,
Response 4: Thank you for your helpful comment. We have clarified the statistical significance of the differences and we have rounded the values to reflect the structured 1–9 scale used in the sensory evaluation. Additionally, we have corrected the formatting of p to be italicized throughout the manuscript.

Round 2
Reviewer 1 Report
Comments and Suggestions for Authors
It is a slight pity that the data collection for this experiment was conducted in 2022, which is now about three years ago. With the social development in recent years, the pursuit of health may have exerted a certain impact on the relevant data to some extent. On the whole, all the comments before were well dressed by authors. This article makes a detailed assessment of the correlation between athletes' preference for high-protein bread and their purchase intention, with comprehensive data analysis.